# Perfectionism, Orthorexia Nervosa, and Body Composition in Young Football Players: A Cross-Sectional Study

**DOI:** 10.3390/nu17030523

**Published:** 2025-01-31

**Authors:** Grzegorz Zydek, Marek Kardas, Wiktoria Staśkiewicz-Bartecka

**Affiliations:** 1Department of Sport Nutrition, Jerzy Kukuczka Academy of Physical Education in Katowice, ul. Mikołowska 72A, 40-065 Katowice, Poland; g.zydek@awf.katowice.pl; 2Department of Food Technology and Quality Evaluation, Faculty of Public Health in Bytom, Medical University of Silesia, 40-055 Katowice, Poland; mkardas@sum.edu.pl

**Keywords:** orthorexia nervosa, body composition, perfectionism, football, soccer, mental health

## Abstract

Background: The pursuit of excellence in sports often drives athletes to maintain rigorous dietary and physical standards, sometimes leading to disordered eating patterns like orthorexia nervosa. The purpose of this study is to evaluate the relationship between perfectionism, body composition, and the risk of orthorexia among young soccer players. Methods: A cross-sectional study was conducted with 93 young football players aged 15–24 from a football academy. Perfectionism was assessed using the Perfectionism in Sport Questionnaire, while orthorexia nervosa risk was evaluated using the Düsseldorf Orthorexia Scale. Body composition parameters, including body mass index, lean body mass, skeletal muscle mass, fat mass, and fat percentage, were analyzed using multi-frequency bioelectrical impedance. Results: Higher levels of negative perfectionism were significantly associated with increased orthorexia nervosa risk (*p* = 0.006), while positive perfectionism showed no significant correlation. Younger players exhibited higher negative perfectionism scores compared to older groups (*p* = 0.043). No significant relationships were found between body mass index, body composition parameters, and orthorexia nervosa risk (*p* > 0.05). Conclusions: This study highlights the psychological underpinnings of orthorexia nervosa risk in young athletes, emphasizing the influence of negative perfectionism and the limited role of body composition. Early intervention focusing on reducing negative perfectionism and promoting adaptive perfectionism could support both psychological well-being and athletic performance. Future research should investigate long-term trends and the role of sociocultural factors in orthorexia nervosa development.

## 1. Introduction

Reaching the highest possible results remains the top priority for modern elite athletes. This entails a drive towards achieving perfection in all facets of athlete preparation to include nutrition and body composition [1,2]. This ambitious goal can complement training and development. However, it can also result in the development of negative attitudes, such as eating disorders (EDs) or extreme perfectionism [3].

The issue of perfection was long debated concerning the Olympic Games. Before the 1976 Olympic Games, the International Olympic Committee rejected the idea of adding a fourth digit to the scoreboards because they felt there was no achieving perfection (score 10.00); however, on the second day of the games, Romania’s 14-year-old gymnast Nadia Comaneci became the first gymnast to earn a perfect 10. She achieved this eight more times during her career, demonstrating that it is indeed possible. From this example, one of the main distinctive features of sport is the desire to be the best, and this is accompanied by constant improvements in skills, which requires time and labor [4].

Perfectionism is a set of beliefs about the necessity of achieving unrealistically high standards, along with focusing on mistakes and fearing failure [5]. In sports, a two-dimensional approach dominates, positing that perfectionistic strivings and perfectionistic concerns are equally important elements of the construct. Setting high personal standards and constantly striving for excellence, while being able to accept discrepancies between reality and the ideal, and to derive satisfaction from one’s achievements, are characteristic traits of perfectionists. Research data and variable descriptions confirm that this is an adaptive characteristic. The second dimension, perfectionistic concerns, encompasses uncertainty about one’s own actions, a sense that something is not going according to expectations and the actual state of affairs, fear of making mistakes and of negative reactions, and ruminating about mistakes that have already been made [6,7]. In recent years, the relationship between perfectionism and selected EDs has been analyzed more frequently, particularly in disciplines where physique and body weight play a critical role [2,8].

Orthorexia nervosa (ON) is a disorder characterized by an excessive preoccupation with healthy eating, manifested through restrictive dietary habits, an obsession with the quality of consumed products, and the exclusion of an increasing number of food groups [9,10]. Although ON has not yet been included in official diagnostic classifications such as the DSM-5 or ICD-11, its impact on athlete functioning can be multifaceted; it has been shown that this disorder is often associated with perfectionism, a sense of control, and the pursuit of excellence, which can exacerbate symptoms and make them more difficult to identify [11,12,13]. ON’s influence on an athlete’s functioning can lead to nutritional deficiencies, excessive mental strain, social isolation, and even severe health complications such as weight loss or a weakened immune system [14,15].

Due to specific psychological and environmental circumstances, ON may develop faster in young athletes. Many factors can contribute to the development of this disorder, including the growing support for “clean eating”, weight control demands, and the influence of social media on eating patterns [16,17]. Additionally, adolescents involved in sports activities often equate healthy eating with athletic success, which increases the likelihood of adopting restrictive eating habits. In this age group, ON can hinder recovery, lower physical performance, and cause long-term health consequences that make it difficult to develop a sports career [18,19]. In light of these risks, it is essential to conduct research that will help us understand the extent to which orthorexia affects the psychophysical health of young athletes and how to effectively identify and minimize the risk of its occurrence.

High-intensity sports, such as soccer, are strongly linked to physical performance and sporting outcomes [8,20]. Optimal muscle mass and an appropriate level of body fat form the foundation of athlete preparation, both factors influencing endurance, speed, and agility [21]. Maintaining the right ratio between muscle mass and body fat may; however, become a subject of excessive focus, especially for young athletes striving for high standards [1]. The pursuit of a “perfect” physique and fears of losing it can foster the development of ON and other unhealthy eating habits. Athletes seeking the “ideal” body experience additional stress because they monitor their body weight, count calories, and compare themselves with other athletes [22]. This can lead to the exclusion of specific food groups due to fears of an increase in body fat, ultimately resulting in a rigid, restrictive lifestyle [2]. In this context, perfectionism becomes crucial. Constructive perfectionism involves a desire for self-improvement and skill development, while its destructive form is characterized by excessive fear of making mistakes and a constant sense of imperfection. Combined with high pressure to achieve an excellent physique, negative perfectionism can lead to obsessive eating behaviors, which in turn increase the risk of EDs such as ON [2,3].

As a result, young soccer players who are continually seeking to improve their athletic performance may fall into the trap of excessive dietary control, which paradoxically lowers their fitness due to nutritional deficiencies, improper meal balance, or recovery issues [14,22]. For this reason, coaches, nutritionists, and athletes themselves must pay attention to maintaining healthy relationships with food while striving for the ideal body. Identifying the negative elements of perfectionism and ON at an early stage is essential for implementing intervention and prevention strategies that will help young athletes to achieve high sports results while maintaining psychological well-being [5,6].

This study aims to explore the relationship between perfectionism, specifically its positive and negative dimensions, and the risk of ON in young football players. Additionally, this study seeks to determine whether body composition parameters, such as muscle mass and fat mass, influence this relationship.

It was hypothesized that athletes who exhibit higher levels of perfectionism have a higher risk of ON, and that athletes with a higher risk of ON will have a different body composition from athletes without risk.

## 2. Materials and Methods

### 2.1. Design and Procedures

This study was conducted in December 2024, involving players from a renowned football academy in the Silesian Agglomeration. All players were examined within two weeks to ensure comparable results. A survey questionnaire was employed for this study, comprising a demographic section as well as the following instruments: the Perfectionism in Sport Questionnaire (PSQ) and the Düsseldorf Orthorexia Scale (PL-DOS), in addition to body composition analysis conducted through Direct Segmental Multi-Frequency Bioelectrical Impedance Analysis (DSM-BIA). The athletes completed the questionnaires and underwent body composition measurements within the same week. The athletes filled out the questionnaires at the academy before training, and instructions were provided beforehand to ensure appropriate comprehension and interpretation of the questions, thereby minimizing errors in completion. The response rate was 100%.

Participants and their legal guardians were informed of this study’s purpose and its anonymity, and they were requested to accept the terms of data sharing. Information regarding the voluntary and informed nature of participation in this study was communicated to and accepted by both guardians and participants before the commencement of this study. This study was conducted in accordance with the Declaration of Helsinki of the World Medical Association. This study received approval from the Bioethics Committee of the Medical University of Silesia in Katowice (approval code: BNW/NWN/0043-3/641/35/23; approval date: 22 September 2023).

### 2.2. Study Participants

This study involved 97 young football players from a football academy. After taking into account the exclusion criteria, 93 players were finally qualified for the analysis. The players were aged between 15 and 24 years, with a mean age of 17.1 ± 1.66. The players were of Polish nationality. The inclusion criteria included (1) consent to participate in this study from the player and/or legal guardian, (2) age of 15 years or older, (3) active player status at the academy at the time of this study, (4) no injury excluding the player from training for at least seven days in the last two months preceding this study, and (5) no contraindications to body composition analysis. The exclusion criteria included (1) incorrectly or incompletely filled-out questionnaire, (2) absence, and (3) presence of contraindications to body composition analysis.

The players were divided into the following age categories:U15–U16 (n = 37; 39.78%);U17–U18 (n = 36; 38.71%);≥U19 (n = 20; 21.51%).

The age groups were chosen as they represent key stages in the physical and psychological development of young athletes. This division allows for the analysis of differences in perfectionism and the risk of orthorexia at various stages of a sports career, as well as an understanding of how age and experience influence these phenomena.

### 2.3. Research Tools

The risk of ON and perfectionism in sport was assessed using a questionnaire consisting of demographic information and two validated scales: the Perfectionism in Sport Questionnaire (PSQ) and the Düsseldorf Orthorexia Scale (PL-DOS). The questionnaire comprised 50 questions, of which 10 were part of the socio-demographic section, 30 came from the PSQ, and 10 from the DOS. The socio-demographic section included general questions about age, playing position, and additional training. Additionally, the demographic section contained a question regarding diagnosed chronic diseases, including mental health disorders such as depression, EDs, and neurosis. None of the respondents reported chronic mental illnesses.

#### 2.3.1. PSQ

Perfectionism was assessed using a Polish-language psychometric instrument, the Perfectionism in Sport Questionnaire (PSQ) [4]. This instrument comprised 30 test items, of which 13 corresponded to the dimension of positive perfectionism (PP) and 17 pertained to negative perfectionism (NP). Each item was evaluated by respondents on a five-point Likert scale, ranging from 1 (“strongly disagree”) to 5 (“strongly agree”) [4].

A semantic analysis facilitated the identification of core components within both scales. For the PP scale, the categories included high personal standards, a focus on improving one’s skills, and the experience of pleasure and satisfaction derived from sports participation. In contrast, the PN scale encapsulated elements such as fear of making mistakes, excessive concern about errors and failures, heightened emotional distress over setbacks, and a demotivating sense of discrepancy between goals and achieved outcomes [4].

The questionnaire was developed and validated by the authors of the scale; the reliability of the tool was determined using Cronbach’s alpha coefficient, yielding a value of 0.91 for the PP scale and 0.93 for the NP scale, indicating a high level of measurement validity and reliability for both dimensions [23].

Based on the PSQ results, a reliability analysis was conducted for PP and NP using McDonald’s omega (ω). The calculated ω value of 0.885 for PP and 0.913 for NP indicates a high level of internal consistency for both subscales.

#### 2.3.2. DOS

The Düsseldorf Orthorexia Scale (DOS) is a screening tool designed to assess orthorexic eating behaviors [24]. The 10-item DOS serves as a unidimensional measure for screening ON, derived from the broader 21-item DOS, which encompasses subscales such as orthorexic eating behaviors, avoidance of additives, and mineral intake [24].

In this study, respondents completed the 10-item DOS, responding on a four-point Likert scale ranging from “strongly disagree” to “strongly agree”, with no reverse-scored items. The maximum possible score was 40 points, and the interpretation of the results followed the established guidelines: a score exceeding 30 indicated the presence of ON, scores between 25 and 29 suggested a risk of ON, while scores below 25 reflected no indication of ON [24]. A Polish adaptation of the DOS (PL-DOS) was used in this study, which was developed and validated by the scale’s authors [25], showing reliability comparable to the original E-DOS, with a Cronbach’s alpha coefficient of 0.84 [24,25].

Based on the results obtained from the PL-DOS, a reliability analysis was conducted using McDonald’s omega (ω). The calculated ω value of 0.726 indicates an acceptable level of reliability of the measurement tool.

#### 2.3.3. Body Composition Analysis

Body composition was assessed using the Direct Segmental Multi-Frequency Bioelectrical Impedance Analysis (DSM-BIA) method with the InBody 770 device (InBody USA, Cerritos, CA, USA). This approach is based on the segmentation of the human body into five cylindrical regions, allowing for the precise measurement of bioelectrical impedance across different body parts, including the trunk, arms, and legs. The system employs tetrapolar, eight-point electrodes to independently measure impedance in each body segment. Measurements are conducted using multiple current frequencies (1 kHz, 5 kHz, 50 kHz, 250 kHz, 500 kHz, and 1000 kHz) and a low current of 80 μA. Each analysis session lasts approximately 60 s and provides detailed data on lean body mass (LBM), total body water (TBW), skeletal muscle mass (SMM), fat mass (FM), and body fat percentage (%FM) [26].

To ensure the consistency and reliability of the measurements, environmental factors were strictly controlled during the analysis. Participants were instructed to avoid intense physical activity and maintain hydration levels for at least 24 h before testing. All measurements were conducted in the morning, following an overnight fast of at least 8 h, to minimize variations due to food and water intake. Room temperature was maintained at a constant level to avoid fluctuations in impedance readings.

Measurements were carried out following the manufacturer’s standardized protocol. To ensure accuracy, the device was calibrated before each testing session using a standard circuit with known impedance values (resistance = 500.0 Ω; reactance = 0.1 Ω; error = 0.9%) [27].

#### 2.3.4. Statistical Analysis

Statistical analyses were performed using Statistica software version 13.3 (StatSoft Poland, Kraków, Poland). The values of the measurable variables were presented as arithmetic means (X) with standard deviation (SD); for non-measurable variables, frequencies (N) and percentages (%) were used.

To examine the normality of the data distribution, the Shapiro–Wilk test was employed. For comparisons between age groups (U15–U16, U17–U18, ≥U19), one-way analysis of variance (ANOVA) was used for continuous variables. Post hoc tests were performed when significant differences were detected.

The relationships between DOS-PL scores, perfectionism dimensions (PP and NP), and body composition variables were assessed using Pearson’s correlation coefficient. Statistical significance was set at *p* < 0.05.

To explore the impact of perfectionism dimensions and age group on DOS-PL scores, a linear regression analysis was conducted. The model included PP and NP as predictors, with DOS-PL scores as the dependent variable. Additionally, group comparisons were included to account for potential differences among age categories. Model fit was assessed using R^2^.

The reliability coefficient, ω McDonald’s, was also calculated to assess the internal consistency of the scales.

A *p*-value of less than 0.05 was considered statistically significant.

## 3. Results

### 3.1. Characteristics and Body Composition of the Study Group

Ninety-three players from a football academy participated in this study, meeting the inclusion and exclusion criteria. Respondents were asked about the occurrence of chronic diseases, and seven individuals provided affirmative responses. The chronic conditions reported by the respondents included atopic dermatitis (n = 1), allergy (n = 1), asthma (n = 4), Hashimoto’s thyroiditis (n = 1), and acne vulgaris (n = 1). They reported regular use of medications such as Symbicort, Euthyrox, Hitaxa, Isotretinoin, Salmex, Astmodil, and Ventolin.

In the question regarding additional training sessions outside the football academy, 83.9% of players reported engaging in additional training, with no differences observed between age groups (*p* = 0.525). Players were also asked about the number of self-conducted training sessions per week. The majority, 50.5% (n = 47), reported 1–2 sessions per week, 16.1% (n = 15) reported 3–4 sessions, 14.0% (n = 13) trained occasionally, and only 3.2% (n = 3) conducted at least one additional session daily. No significant differences in the frequency of additional training were observed across the analyzed age groups (*p* = 0.489).

Regarding on-field positions, goalkeepers accounted for 10.8% (n = 10) of the participants, forwards 21.5% (n = 20), defenders 26.9% (n = 25), and midfielders 40.9% (n = 38). The characteristics of the study group and the body composition of the players by age groups are presented in Table 1.

### 3.2. Assessment of the Risk of ON

Based on the DOS-PL questionnaire, increased risk of ON was found in 18.3% (n = 17) of players, and the occurrence of ON behaviors was found in 9.7% (n = 9) of respondents. No significant relationships were found between the interpretation and the number of points obtained on the DOS-PL scale across the analyzed age groups (*p* = 0.420; *p* = 0.611). Details are shown in Table 2.

The values obtained from the body composition analysis were examined in the context of ON risk. No significant relationships were identified between the body composition parameters and the occurrence of ON. Detailed results are presented in Table 3.

### 3.3. Evaluation of Perfectionism

This study analyzed the results obtained in the PSQ to positive and negative perfectionism. Analysis of the results showed statistically significant differences between the age groups in the number of points obtained in relation to NP (*p* = 0.43). No differences were shown concerning PP (*p* = 0.338). This may suggest that older athletes are less affected by perfectionism’s negative aspects than younger athletes. The results are shown in Table 4.

### 3.4. Correlation Between ON Risk, Positive Perfectionism, Negative Perfectionism, and Body Composition

The results of the correlation analysis indicate several significant relationships between the variables. Positive perfectionism shows a weak, non-significant correlation with DOS-PL points (r = 0.164, *p* = 0.116). In contrast, negative perfectionism demonstrates a significant correlation with DOS-PL points (r = 0.283, *p* = 0.006), suggesting that more negative aspects of perfectionism may be associated with experiencing negative pressure in sports. Regarding body parameters, BMI does not show any significant correlations with either type of perfectionism or DOS-PL points, similarly to individual body composition parameters. Detailed information is presented in Table 5.

The regression analysis results indicate a significant impact of NP on DOS-PL points, as confirmed by the t-value (t = 2.907) and *p*-value (*p* = 0.005). PP did not reach statistical significance (estimate, *p* = 0.053). Comparisons between age groups showed no significant differences, both for the U15–U16 group compared to U17–U18 (*p* = 0.781) and for the U19 < group compared to U17–U18 (*p* = 0.820). Model fit measures, including an R^2^ value of 0.122, suggest moderate explanatory power of the independent variables for the variability in DOS-PL points (Table 6).

The charts illustrate the relationship between DOS-PL points and NP and PP across age groups (U15-U16, U17-U18, ≥U19) (Figure 1).

## 4. Discussion

This study provides significant insights into the interrelationships between psychological traits and physical parameters in young athletes [28,29,30,31,32]. The analysis of perfectionism and its association with the risk of ON aims to highlight the dynamics of striving for excellence in sports and the potential psychological costs associated with it [10,33,34,35,36].

A significant correlation was identified between NP and scores on the DOS-PL scale, which assesses the risk of ON. NP represents a critical risk factor in the development of EDs. It is associated with excessive fear of mistakes, inadequacy, and a strong focus on failures, which can lead to obsessive eating control and restrictive dietary patterns. In contrast, PP, understood as striving for high standards while maintaining flexibility and accepting errors, showed no significant correlation with the risk of ON development. This confirms prior findings suggesting that the primary risk factor for EDs is not merely the pursuit of excellence but the burdensome belief in the necessity of being “perfect”.

An interesting aspect of this study is the lack of significant relationships between body composition parameters, including BMI, muscle mass, or fat percentage, and the risk of ON. This result may suggest that ON risk is more strongly influenced by psychological factors such as perfectionism rather than physical parameters [37,38,39]. This reflects the complexity of EDs, which, as studies show, are more related to the perception of one’s own body than its actual state. Research by Staśkiewicz-Bartecka et al. [10] on a group of elite athletes in Poland similarly demonstrated no correlation between body composition and ON risk, though it did reveal an association between the EAT-26 questionnaire results and ED risk. Individuals at risk of ON often focus on potential dietary errors and overly restrictive diet rules. In this sense, food quality (perceived as “ideal purity” by ON sufferers) matters more than quantity, which may explain the lack of correlation with body composition parameters.

The findings indicating significant differences in NP levels between younger athletes (e.g., U15–U16) and older athletes (U19) may stem from developmental and environmental factors. Younger athletes are at an earlier stage of forming their athletic identity and are still learning effective ways to cope with pressure. They also tend to be more susceptible to external evaluation, which may translate into higher levels of fear of mistakes and excessive self-criticism. The ability to accept imperfections and cope with sports failures generally increases with age and experience. The development of cognitive skills, emotional regulation, and resilience also contribute to this process. Older athletes are better able to manage their expectations regarding their performance, distinguishing their ambitions and striving for high standards from the destructive fear of failure [40,41]. As a result, adaptive perfectionism (PP) or at least lower levels of NP may prevail among them.

The context of youth sports also underscores the significant role of parents and coaches, and sometimes peer pressure, in fostering NP in younger age groups. This is particularly true in environments where minor mistakes are harshly criticized and unrealistic standards are set. Some young athletes develop adaptive ways of thinking as they gain a better understanding of their abilities, change their training styles, or exchange experiences with other athletes. This reduces NP levels and the associated anxiety [28,29,30,31,32].

Both NP and PP may have specific origins in sports. The sports environment, especially in youth groups, plays a critical role in shaping attitudes toward competition [42]. Young athletes’ sensitivity to external evaluation, such as from coaches, parents, and peers, can lead to a pursuit of “perfection” and excessive focus on mistakes. In contrast, older athletes, with more experience and stress-management skills, may learn a more adaptive approach to their flaws. Cognitive maturity and emotional regulation mechanisms reduce their tendency toward extreme forms of perfectionism [28,29,30,31,32,33].

In the field of self-determination theory, studies have shown that the motivational environment surrounding a young athlete influences how they perceive their achievements and potential failures. Athletes are encouraged to develop individual skills and learn from mistakes in teams or clubs that adopt a developmental approach, known as the “mastery climate”. In such situations, perfectionism may take on a more adaptive character (PP). However, in scenarios where excessive emphasis is placed on outcomes and social comparisons, there is a greater risk of NP and related behavioral patterns [10].

The clear lack of correlation between ON risk and body composition parameters confirms that psychological beliefs (such as fear of mistakes or the perception of dietary “purity”) have a more significant influence on eating habits than objective measures of body weight or fat percentage.

This study’s findings underscore the importance of psychological support and education for young athletes, coaches, and parents in preventing the development of NP and the risk of EDs (including ON). It is therefore essential to distinguish between adaptive striving for high standards and destructive fear of mistakes. In sports, it is crucial to create an environment that encourages self-improvement, where mistakes are seen as a natural part of skill development [39]. A multifaceted understanding of the social and cultural context (such as peer pressure, coach demands, and media influence) will be of interest for further research. Longitudinal studies will also be valuable for tracking how perfectionism and body perception change with age and athletic experience.

The age group differentiation was an essential aspect of this study as it allowed for an examination of the dynamics of perfectionism and its relationship with ON risk across different stages of athlete development [10]. The use of standardized measurement tools ensures that the results are reliable and comparable. An additional advantage is the combination of physical parameter analysis with psychological dimensions (perfectionism), which broadens the interpretative perspective and allows for a more comprehensive approach to the issue of EDs in sports.

This study is mainly limited by its cross-sectional nature, which prevents drawing causal conclusions. Generalizing the results to other populations may be challenging as this study involved specific groups of youth athletes. Additionally, the reliance on self-reported questionnaires may generate social desirability bias and some distortions in reporting eating behaviors and personality traits. Another limitation is the relatively small sample size, which might reduce the statistical power of the findings and limit the robustness of the conclusions.

## 5. Conclusions

This study confirmed that negative perfectionism is significantly associated with a higher risk of ON among young football players, while positive perfectionism showed no significant correlation with ON risk. Furthermore, the analysis revealed that body composition parameters, such as muscle mass and fat mass, do not differ significantly among athletes with varying levels of perfectionism and do not moderate the relationship between perfectionism and ON risk. The findings emphasize that psychological traits, such as perfectionism, have a greater influence on the development of ON risk than physical indicators.

The results highlight the need to implement educational programs and psychological support for young athletes, their coaches, and parents. Special attention should be given to identifying and reducing negative perfectionism while promoting adaptive forms of perfectionism that support athletic development without negatively affecting mental health. Creating a sports environment that fosters self-improvement, treats mistakes as a natural part of growth, and eliminates excessive emphasis on outcomes is crucial for protecting young athletes from ON and its psychological consequences.

## Figures and Tables

**Figure 1 nutrients-17-00523-f001:**
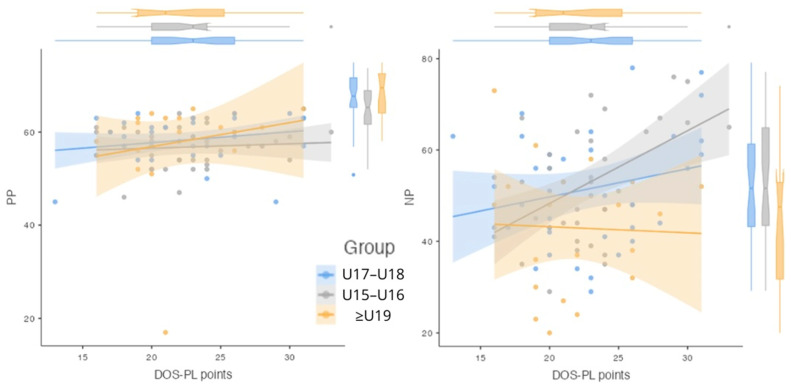
Scatter plot of DOS-PL points and PP (positive perfectionism) and NP (negative perfectionism) by age group (n = 93).

**Table 1 nutrients-17-00523-t001:** The characteristics and body composition of athletes considering age groups (n = 93).

Variable	Total (n = 93) X ± SD	U15–U16 (n = 37) X ± SD	U17–U18 (n = 36) X ± SD	U19≤ (n = 20) X ± SD	*p*-Value
Age [years]	17.09 ± 1.66	15.66 ± 0.42	17.17 ± 0.59	19.57 ± 1.40	0.001 **
Body mass [kg]	66.48 ± 10.3	60.60 ± 7.33	67.93 ± 8.84	72.15 ± 11.05	0.001 **
Height [cm]	178.32 ± 7.09	174.89 ± 6.08	179.28 ± 6.44	182.93 ± 7.08	0.001 **
BMI [kg/m^2^]	20.79 ± 2.05	19.75 ± 1.64	21.06 ± 1.89	22.24 ± 2.17	0.001 **
LBM [kg]	60.43 ± 9.16	56.82 ± 5.68	61.12 ± 9.44	65.85 ± 11.11	0.002 *
TBW [kg]	44.27 ± 6.63	41.69 ± 4.11	44.74 ± 6.83	48.19 ± 8.06	0.002 *
SMM [kg]	34.19 ± 5.55	31.94 ± 3.44	34.67 ± 5.74	37.49 ± 6.66	0.001 **
FM [kg]	6.05 ± 2.29	6.07 ± 2.17	5.86 ± 2.38	6.38 ± 2.42	0.743
%FM [%]	9.01 ± 2.79	9.52 ± 2.86	8.60 ± 2.69	8.79 ± 2.83	0.357

* = *p* < 0.05, ** = *p* < 0.01; X: mean; SD: standard deviation; BMI: body mass index; LBM: lean body mass; TBW: total body water; SMM: skeletal muscle mass; FM: fat mass; %FM: fat mass percentage.

**Table 2 nutrients-17-00523-t002:** ON risk assessment and DOS-PL scores by age group (n = 93).

DOS-PL—Interpretation				
Group	U15–U16 n = 37	U17–U18 n = 36	U19< n = 20	Total n = 93
No risk n (%)	30 (81.1)	23 (63.9)	14 (70.0)	67 (72.0)
Presence n (%)	3 (8.1)	5 (13.9)	1 (5.0)	9 (9.7)
Risk of ON n (%)	4 (10.8)	8 (22.2)	5 (25.5)	17 (18.3)
*p*-value	0.420			
DOS-PL points X ± SD	22.59 ± 3.98	23.00 ± 4.70	21.80 ± 4.09	22.58 ± 4.27
*p*-value	0.611			

X—mean; SD—standard deviation.

**Table 3 nutrients-17-00523-t003:** Relationships between body composition parameters and ON risk.

Variable	No Risk X ± SD	Risk of ON X ± SD	Presence X ± SD	*p*-Value
Age [years]	17.09 ± 1.79	17.28 ± 1.35	16.67 ± 1.23	0.515
Body mass [kg]	66.58 ± 11.24	65.78 ± 8.13	67.06 ± 6.132	0.902
Height [cm]	178.46 ± 7.58	177.82 ± 5.87	177.22 ± 5.84	0.935
BMI [kg/m^2^]	20.77 ± 2.22	20.74 ± 1.86	21.06 ± 0.78	0.699
LBM [kg]	60.11 ± 9.08	61.19 ± 9.07	61.30 ± 10.80	0.883
TBW [kg]	44.06 ± 6.58	44.75 ± 6.57	44.89 ± 7.78	0.902
SMM [kg]	33.99 ± 5.51	34.75 ± 5.46	34.64 ± 6.56	0.862
FM [kg]	6.00 ± 2.26	6.26 ± 2.33	6.09 ± 2.76	0.915
%FM [%]	8.98 ± 2.87	9.19 ± 2.54	8.88 ± 2.88	0.948

X: mean; SD: standard deviation; BMI: body mass index; LBM: lean body mass; TBW: total body water; SMM: skeletal muscle mass; FM: fat mass; %FM: fat mass percentage.

**Table 4 nutrients-17-00523-t004:** Number of PSQ scores by age group (n = 93).

PSQ Points				
Group	U15–U16 n = 37	U17–U18 n = 36	U19< n = 20	Total n = 93
PP X ± SD	56.78 ± 4.38	58.42 ± 4.91	57.80 ± 10.56	57.63 ± 6.31
*p*-value	0.338			
NP X ± SD	52.43 ± 12.34	51.58 ± 12.56	42.95 ± 14.22	50.06 ± 13.3
*p*-value	0.043 *			

* = *p* < 0.05; X—mean; SD—standard deviation; PP—positive perfectionism; NP—negative perfectionism.

**Table 5 nutrients-17-00523-t005:** Correlation matrix between DOS-PL, PP and NP points, BMI, and body composition parameters (n = 93).

		DOS-PL Points	PP Points	NP Points	BMI [kg/m^2^]	TBW [kg]	FM [kg]	LBM [kg]	SMM [kg]	%FM [%]
Faculty of Public Health	r Pearsona	—								
	*df*	—								
	*p*	—								
	95% upper CI	—								
	95% lower CI	—								
Faculty of Public Health	r Pearsona	0.164	—							
	*df*	91	—							
	*p*	0.116	—							
	95% upper CI	0.356	—							
	95% lower CI	−0.041	—							
Faculty of Public Health	r Pearsona	0.283 **	−0.124	—						
	*df*	91	91	—						
	*p*	0.006	0.238	—						
	95% upper CI	0.461	0.082	—						
	95% lower CI	0.085	−0.319	—						
Faculty of Public Health	r Pearsona	−0.019	0.071	0.016	—					
	*df*	91	91	91	—					
	*p*	0.856	0.497	0.878	—					
	95% upper CI	0.185	0.271	0.219	—					
	95% lower CI	−0.222	−0.134	−0.188	—					
Faculty of Public Health	r Pearsona	0.032	0.039	−0.025	0.446 ***	—				
	*df*	91	91	91	91	—				
	*p*	0.759	0.713	0.810	<0.001	—				
	95% upper CI	0.234	0.241	0.179	0.596	—				
	95% lower CI	−0.173	−0.166	−0.228	0.267	—				
Faculty of Public Health	r Pearsona	0.105	0.083	−0.034	0.218 *	0.393 ***	—			
	*df*	91	91	91	91	91	—			
	*p*	0.316	0.431	0.749	0.035	<0.001	—			
	95% upper CI	0.302	0.282	0.171	0.404	0.553	—			
	95% lower CI	−0.101	−0.123	−0.236	0.015	0.206	—			
Faculty of Public Health	r Pearsona	0.034	0.040	−0.026	0.448 ***	1.000 ***	0.395 ***	—		
	*df*	91	91	91	91	91	91	—		
	*p*	0.746	0.704	0.807	<0.001	<0.001	<0.001	—		
	95% upper CI	0.236	0.242	0.179	0.598	1.000	0.554	—		
	95% lower CI	−0.171	−0.165	−0.228	0.269	1.000	0.208	—		
SMM [kg]	r Pearsona	0.027	0.045	−0.031	0.451 ***	0.998 ***	0.383 ***	0.999 ***	—	
	*df*	91	91	91	91	91	91	91	—	
	*p*	0.794	0.666	0.771	<0.001	<0.001	<0.001	<0.001	—	
	95% upper CI	0.230	0.247	0.174	0.600	0.999	0.544	0.999	—	
	95% lower CI	−0.177	−0.160	−0.233	0.272	0.997	0.194	0.998	—	
%FM [%]	r Pearsona	0.091	0.052	−0.039	0.079	0.006	0.909 ***	0.007	−0.004	—
	*df*	91	91	91	91	91	91	91	91	—
	*p*	0.387	0.624	0.708	0.454	0.955	<0.001	0.945	0.967	—
	95% upper CI	0.289	0.253	0.166	0.278	0.209	0.939	0.211	0.199	—
	95% lower CI	−0.115	−0.154	−0.241	−0.127	−0.198	0.865	−0.197	−0.208	—

PP—positive perfectionism; NP—negative perfectionism; * = *p* < 0.05, ** = *p* < 0.01, *** = *p* < 0.001; *df*—degrees of freedom; CI—confidence interval; BMI: body mass index; LBM: lean body mass; TBW: total body water; SMM: skeletal muscle mass; FM: fat mass; %FM: fat mass percentage.

**Table 6 nutrients-17-00523-t006:** Linear regression analysis of DOS-PL, PP, and NP scores by age group (N = 133).

Model Coefficients—DOS-PL Points; R = 0.349; R^2^ = 0.122						
Predictor	Estimate	SE	95% CI		t	*p*-Value
			Lower	Upper		
Intercept	10.071	4.614	0.901	19.241	2.183	0.032 *
PP	0.135	0.069	−0.002	0.271	1.961	0.053
NP	0.098	0.033	0.031	0.165	2.907	0.005 **
Group						
U15–U16–U17–U18	−0.269	0.965	−2.186	1.649	−0.279	0.781
≥U19–U17–U18	−0.270	1.180	−2.615	2.076	−0.229	0.820

PP—positive perfectionism; NP—negative perfectionism; SE—standard error; CI—confidence interval; t—t-value; * = *p* < 0.05, ** = *p* < 0.01.

## Data Availability

The raw data supporting the conclusions of this article will be made available by the authors upon request.

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
