# Peer review of "Perfectionism, Orthorexia Nervosa, and Body Composition in Young Football Players: A Cross-Sectional Study"

_nutrients, 2025, doi:10.3390/nu17030523_

Round 1

Reviewer 1 Report

Comments and Suggestions for Authors

The manuscript entitled “Perfectionism, Orthorexia Nervosa, and Body Composition in Young Football Players: A Cross-Sectional Study” is very interesting with well-written. Some problems need to be addressed as follows.

1.       the Perfectionism in Sport Questionnaire (PSQ) and the Düsseldorf Orthorexia Scale (PL-DOS)” Please provide the questionnaires of reliability and validity values in your study; or add relevant references to support your idea.

2.       The notes of Tables, X – Average should be changed to X-Mean.

3.       PU - Confidence Interval should be changed to “CI- Confidence Interval” in the Table 5.

4.       The decimal places in table 6 are either the same as 2 or 3 decimal places in the previous tables. Please consider it.

5.       Line 365, “Cognitive maturity and emotional regulation mechanisms reduce their tendency toward extreme forms of perfectionism” please add reference.

6.       The small sample size of the study is also a limitation, please point out in the discussion section.

7.       The conclusion section needs to describe important conclusions based on the study results.

Author Response

Thank you so much for taking the time to evaluate our work. We have tried to incorporate all your valuable suggestions. If we could improve our work in any way, please let us know.

The manuscript entitled “Perfectionism, Orthorexia Nervosa, and Body Composition in Young Football Players: A Cross-Sectional Study” is very interesting with well-written. Some problems need to be addressed as follows.

Comment 1

  1. “the Perfectionism in Sport Questionnaire (PSQ) and the Düsseldorf Orthorexia Scale (PL-DOS)” Please provide the questionnaires of reliability and validity values in your study; or add relevant references to support your idea.

McDonald’s omega was calculated for both scales, as described in detail in the methodology section, in lines 180–182 and 196–198. For the PSQ, the values were 0.885 for Positive Perfectionism (PP) and 0.913 for Negative Perfectionism (NP), while for the PL-DOS, the value was 0.726. Both questionnaires were developed and validated in Polish, and their validity was confirmed in the studies by Waleriańczyk and Stolarski (2016) and Brytek-Matera (2021). These references are indicated in the methodology section, in lines 176–179 and 196–198.

Comment 2

  1. The notes of Tables, X – Average should be changed to X-Mean.

Thank you very much for your suggestion, corrected.

Comment 3

  1. PU - Confidence Interval should be changed to “CI- Confidence Interval” in the Table 5.

Thank you very much for your suggestion, corrected.

Comment 4

  1. The decimal places in table 6 are either the same as 2 or 3 decimal places in the previous tables. Please consider it.

We have corrected the number of decimal places. All means and standard deviations are now reported to two decimal places, while more precise statistics are presented to three decimal places.

Comment 5

  1. Line 365, “Cognitive maturity and emotional regulation mechanisms reduce their tendency toward extreme forms of perfectionism” please add reference.

Citation provided.

  1. Hill, A. P.; Curran, T. Multidimensional Perfectionism and Burnout: A Meta-Analysis. Pers. Soc. Psychol. Rev. 2016, 20 (3), 269–288.

Comment 6

  1. The small sample size of the study is also a limitation, please point out in the discussion section.

Thank you very much for your suggestion, corrected.

Comment 7

  1. The conclusion section needs to describe important conclusions based on the study results.

Thank you very much for your valuable comment, corrected

Thank you for your help. Your guidance is invaluable.

Kind regards,

Authors.

Reviewer 2 Report

Comments and Suggestions for Authors

The article entitled “Perfectionism, orthorexia nervosa and body composition in young soccer players: a cross-sectional study”, brings together the characteristics of a cross-sectional descriptive empirical research, with the only drawback that the sample is too small. However, since it is aimed at young footballers from non-professional clubs in the same country, I understand that it is enough.

The summary contains the elementary parts of a scientific article, although the objective does not appear, which must always be in this part of the research so that the reader can navigate without problems.

Regarding the development, the introduction is sufficiently broad, contains 23 quotes and defines the concepts clearly. However, although it sets objectives, they are not clear because they are not called as such. They say:

“The present study seeks to investigate”: 1. the relationship between perfectionism (to cover both its positive and negative dimensions) and the risk of Orthorexia Nervosa (ON) among young soccer players. Additionally, 2. aims to elucidate how specific indices of body composition, such as muscle mass and fat mass, may moderate this association. More precisely, 3. the analysis will examine whether body composition parameters differ between athletes with different levels of perfectionism and whether higher levels of negative perfectionism are associated with a higher risk of ON.

They then propose two hypotheses, which in my opinion are not necessary for a study of these characteristics:

1. Athletes who exhibit higher levels of negative perfectionism have a higher risk of ON.

2. these trends would be reflected in measurable differences in body composition.

In material and method, the instruments used are appropriate: The Perfectionism in Sport Questionnaire (PSQ) and the Düsseldorf Orthorexia Scale (PL-DOS). Body composition, as well as sociodemographic variables, also contain sufficient data to reinforce the study.

The statistical study of the results is correct. The 6 Tables they present, as well as the 2 figures, are well explained and help understanding.

The discussion is carried out using the initial data and the results achieved, I think it is sufficient and well written.

The conclusions respond to the stated objectives.

Regarding the references, there are 47, most of them from the last 10 years, which gives the work important updating support.

Author Response

Thank you so much for taking the time to evaluate our work. We have tried to incorporate all your valuable suggestions. If we could improve our work in any way, please let us know.

The manuscript entitled “Perfectionism, Orthorexia Nervosa, and Body Composition in Young Football Players: A Cross-Sectional Study” is very interesting with well-written. Some problems need to be addressed as follows.

Comment 1

The summary contains the elementary parts of a scientific article, although the objective does not appear, which must always be in this part of the research so that the reader can navigate without problems..

Thank you very much for your suggestion, corrected.

Comment 2

Regarding the development, the introduction is sufficiently broad, contains 23 quotes and defines the concepts clearly. However, although it sets objectives, they are not clear because they are not called as such. They say:

“The present study seeks to investigate”: 1. the relationship between perfectionism (to cover both its positive and negative dimensions) and the risk of Orthorexia Nervosa (ON) among young soccer players. Additionally, 2. aims to elucidate how specific indices of body composition, such as muscle mass and fat mass, may moderate this association. More precisely, 3. the analysis will examine whether body composition parameters differ between athletes with different levels of perfectionism and whether higher levels of negative perfectionism are associated with a higher risk of ON.

They then propose two hypotheses, which in my opinion are not necessary for a study of these characteristics:

  1. Athletes who exhibit higher levels of negative perfectionism have a higher risk of ON.

  1. these trends would be reflected in measurable differences in body composition..

Thank you very much for your valuable comment, corrected

Comment 3

In material and method, the instruments used are appropriate: The Perfectionism in Sport Questionnaire (PSQ) and the Düsseldorf Orthorexia Scale (PL-DOS). Body composition, as well as sociodemographic variables, also contain sufficient data to reinforce the study.

The statistical study of the results is correct. The 6 Tables they present, as well as the 2 figures, are well explained and help understanding.

The discussion is carried out using the initial data and the results achieved, I think it is sufficient and well written.

The conclusions respond to the stated objectives.

Regarding the references, there are 47, most of them from the last 10 years, which gives the work important updating support.

Thank you for your help. Your guidance is invaluable.

Kind regards,

Authors.